# Suppressing the Spikes in Electroencephalogram via an Iterative Joint Singular Spectrum Analysis and Low-Rank Decomposition Approach

**DOI:** 10.3390/s20020341

**Published:** 2020-01-07

**Authors:** Zikang Tian, Bingo Wing-Kuen Ling, Xueling Zhou, Ringo Wai-Kit Lam, Kok-Lay Teo

**Affiliations:** 1School of Information Engineering, Guangdong University of Technology, Guangzhou 510006, China; Zikang_Chio_Tian@163.com (Z.T.); zzhouxueling@163.com (X.Z.); 2AI Mnemonic Limited, Science Park, Hong Kong, China; ringolam.hk@gmail.com; 3School of Electrical Engineering, Computing and Mathematical Sciences, Curtin University, Perth WA 6845, Australia; k.l.teo@curtin.edu.au

**Keywords:** suppressing the spikes, electroencephalogram, singular spectrum analysis, low-rank decomposition

## Abstract

The novelty and the contribution of this paper consists of applying an iterative joint singular spectrum analysis and low-rank decomposition approach for suppressing the spikes in an electroencephalogram. First, an electroencephalogram is filtered by an ideal lowpass filter via removing its discrete Fourier transform coefficients outside the δ wave band, the θ wave band, the α wave band, the β wave band and the γ wave band. Second, the singular spectrum analysis is performed on the filtered electroencephalogram to obtain the singular spectrum analysis components. The singular spectrum analysis components are sorted according to the magnitudes of their corresponding eigenvalues. The singular spectrum analysis components are sequentially added together starting from the last singular spectrum analysis component. If the variance of the summed singular spectrum analysis component under the unit energy normalization is larger than a threshold value, then the summation is terminated. The summed singular spectrum analysis component forms the first scale of the electroencephalogram. The rest singular spectrum analysis components are also summed up together separately to form the residue of the electroencephalogram. Next, the low-rank decomposition is performed on the residue of the electroencephalogram to obtain both the low-rank component and the sparse component. The low-rank component is added to the previous scale of the electroencephalogram to obtain the next scale of the electroencephalogram. Finally, the above procedures are repeated on the sparse component until the variance of the current scale of the electroencephalogram under the unit energy normalization is larger than another threshold value. The computer numerical simulation results show that the spike suppression performance based on our proposed method outperforms that based on the state-of-the-art methods.

## 1. Introduction

Electroencephalogram is a brain signal that reflects the activities of the human body such as the splendid, the sleeping quality, the emotion level and the epileptic disorder. By studying the electroencephalogram, many related diseases can be diagnosed effectively [1,2,3]. However, the quality of the acquired signal is usually very poor—this may be due to the sensor limitations, the background environmental noise and the human factors. Here, human factors include human motions such as the winks. It is found that the electroencephalogram is usually corrupted by the spikes [4,5,6]. Hence, the suppression of spikes plays an important role in the medical community.

There are many existing linear and nonadaptive time frequency analysis-based methods for suppressing the noise in a signal. If the noise is wide sense stationary, then the power spectral density of the denoised signal is equal to that of the original signal multiplied to the squares of the magnitude response of the filter. Therefore, the conventional filtering approach based on a linear time invariant filter is employed to suppress the noise [7,8,9]. Nevertheless, the noise is not wide sense stationary in the practical situation. To address this difficulty, the filter bank and wavelets approach is employed. A filter bank refers to a system consisting of two sub-systems, namely the analysis bank and the synthesis bank. The analysis bank consists of a bank of filters and a set of downsamplers. Since the input signal is decomposed into different subband components via the filters with different frequency bands, the input signal can be analyzed via the analysis bank. The soft thresholding is applied to the subband coefficients. The synthesis bank consists of a bank of filters and a set of upsamplers. Since the denoised subband components are combined via the filters, the denoised signal can be synthesized via the synthesis bank [10,11,12]. Since the downsamplers and the upsamplers are linear time periodically varying systems, this approach is effective to cyclostationary noise [10,11,12]. Nevertheless, as both the analysis filters and the synthesis filters are required to be predefined prior, the denoising performance is highly dependent on the choice of the filters.

To address the above difficulty, many nonlinear and adaptive time frequency analysis-based methods are proposed for suppressing the noise in a signal. For example, the empirical mode decomposition is employed to decompose a signal into a set of intrinsic mode functions. The first intrinsic mode function is discarded, and the rest of the intrinsic mode functions are summed up together to obtain the denoised signal [13,14]. Although this approach is effective for the additive white Gaussian noise, it is not effective for the additive white impulsive noise. This is because the first intrinsic mode function has the highest total number of the extrema, while the total number of the extrema of the impulses is very small. As the spikes behave like the impulses, this approach is not effective for suppressing the spikes in the electroencephalogram.

Likewise, the singular spectrum analysis is also employed to decompose a signal into a set of singular spectrum analysis components. However, for the conventional singular spectrum analysis approach, the singular spectrum analysis components corresponding to the small eigenvalues are discarded because they do not contribute significantly to the signal components. On the other hand, the singular spectrum analysis components corresponding to the large eigenvalues are used for the reconstruction of the signal. Hence, they are summed up together to obtain the denoised signal [15,16,17]. However, as the spikes are with the large magnitudes and the singular spectrum analysis components corresponding to the spikes are usually corresponding to the large eigenvalues, the spikes are still corrupted in the denoised signal.

To tackle the above issue, the low-rank decomposition is employed for decomposing a signal into two components, namely the signal component and the noise component. Here, the decomposition problem is formulated as an optimization problem such that the weighted sum of the rank of the signal component and the total number of the nonzero elements in the noise component is minimized [18,19,20]. Although this approach is effective for suppressing the impulsive noise, it is not effective for suppressing the Gaussian noise. Nevertheless, the signals acquired in the practical situation are corrupted by both the impulsive noise and the Gaussian noise. Hence, this approach does not yield a desirable denoising performance.

Since different approaches have their own advantages and disadvantages, this paper proposes an iterative joint singular spectrum analysis and low-rank decomposition approach for suppressing the spikes in an electroencephalogram. Unlike the conventional singular spectrum analysis based denoising methods [15,16,17], this paper sums up the singular spectrum analysis components corresponding to the small eigenvalues. Also, this paper does not completely discard the sparse component obtained by the low-rank decomposition. On the other hand, the singular spectrum analysis is performed on the sparse component again and the useful information in the sparse component is dug out via an iterative approach. The outline of this paper is as follows. Section 2 reviews both the singular spectrum analysis and the low-rank decomposition. Section 3 presents the proposed spike suppression method. The computer numerical simulation results are presented in Section 4. Finally, a conclusion is drawn in Section 5.

## 2. Reviews on the Singular Spectrum Analysis and the Low-Rank Decomposition

### 2.1. Review on the Singular Spectrum Analysis [15,16,17]

The singular spectrum analysis is to represent a signal as the sum of the singular spectrum analysis components. The procedures for performing the singular spectrum analysis are as follows. Let N be the length of a signal and the vector form representation of the signal be $x={\left[\begin{array}{ccc} x_{1} & \cdots & x_{N} \end{array}\right]}^{T}$. Denote L as the window length. Here, $L\leq \frac{N}{2}$. Define K=N−L+1. Define $X_{k}={\left[\begin{array}{ccc} x_{k} & \cdots & x_{k+L-1} \end{array}\right]}^{T}$ for k=1,⋯,K. The first step is to construct the trajectory matrix as $X=\left[\begin{array}{ccc} X_{1} & \cdots & X_{K} \end{array}\right]$. That is:(1)$$X=\left[\begin{array}{cccc} x_{1} & x_{2} & \cdots & x_{N-L+1}\\ x_{2} & x_{3} & \cdots & x_{N-L+2}\\ \vdots & \vdots & \ddots & \vdots \\ x_{L} & x_{L+1} & \cdots & x_{N} \end{array}\right].$$

It is worth noting that X is a Hankel matrix. The second step is to apply the singular value decomposition to $XX^{T}$. Denote $\lambda_{l}$ for l=1,⋯,L as the eigenvalues of $XX^{T}$. Here, it is assumed that they are sorted in the decreasing order. That is, $\lambda_{1}\geq \ldots \geq \lambda_{L}\geq 0$. Denote $\Lambda =diag\left({\left[\begin{array}{ccc} \lambda_{1} & \cdots & \lambda_{L} \end{array}\right]}^{T}\right)$. Let $U_{l}$ for l=1,⋯,L be the eigenvectors of X. Define $U=\left[\begin{array}{ccc} U_{1} & \cdots & U_{L} \end{array}\right]$. Then, we have $XX^{T}=U\Lambda U^{T}$. Define $V_{l}=\frac{X^{T}U_{l}}{\sqrt{\lambda_{l}}}$ for l=1,⋯,L. It can be shown that the trajectory matrix can be written as:(2)$$X=\sum_{l=1}^{L}\sqrt{\lambda_{l}}U_{l}V_{l}^{T}=\sum_{l=1}^{L}{\tilde{X}}_{l}.$$

The third step is to group ${\tilde{X}}_{l}$ for l=1,⋯,L based on a certain criterion. Assume that the index set {1,2…,L} is partitioned into $\tilde{M}$ disjoint subsets and they are denoted as $I_{m}$ for $m=1,\cdots ,\tilde{M}$. Let ${\hat{X}}_{m}=\sum_{i\in I_{m}}{\tilde{X}}_{i}$ for $m=1,\cdots ,\tilde{M}$. It can be shown that X can be represented as:(3)$$X=\sum_{m=1}^{\tilde{M}}{\hat{X}}_{m}.$$

The fourth step is to perform the de-Hankelization on ${\hat{X}}_{m}$ for $m=1,\cdots ,\tilde{M}$. This is to convert ${\hat{X}}_{m}$ to a one-dimensional signal for $m=1,\cdots ,\tilde{M}$. In particular, let ${\hat{x}}_{a,b,m}$ be the element in the $a^{\mathrm{th}}$ row and the $b^{\mathrm{th}}$ column of ${\hat{X}}_{m}$ for a=1,⋯,L, for b=1,⋯,K and for $m=1,\cdots ,\tilde{M}$. Define:(4)$${\bar{x}}_{n,m}=\{\begin{array}{c} \frac{1}{n}\sum_{p=1}^{n}{\hat{x}}_{p,n-p+1,m}\;for\;1\leq n<L\\ \frac{1}{L}\sum_{p=1}^{L}{\hat{x}}_{p,n-p+1,m}\;for\;L\leq n<K\\ \frac{1}{N-n+1}\sum_{p=n-K+1}^{L}{\hat{x}}_{p,n-p+1,m}\;for\;K\leq n\leq N \end{array}.$$
for $m=1,\cdots ,\tilde{M}$ and for n=1,⋯,N. Here, ${\bar{x}}_{n,m}$ is obtained by averaging ${\hat{x}}_{a,b,m}$ in the $n^{\mathrm{th}}$ off-diagonal of ${\hat{X}}_{m}$ for n=1,⋯,N and for $m=1,\cdots ,\tilde{M}$. Define the singular spectrum analysis component as ${\bar{x}}_{m}={\left[\begin{array}{ccc} {\bar{x}}_{1,m} & \cdots & {\bar{x}}_{N,m} \end{array}\right]}^{T}$ for $m=1,\cdots ,\tilde{M}$.

Since the singular spectrum analysis components are obtained based on the magnitudes of the eigenvalues of the trajectory matrix, the singular spectrum analysis is a magnitude-based signal decomposition method. Therefore, there is not a simple relationship between the indices and the frequency bands of the singular spectrum analysis components. As a result, the frequency bands of the singular spectrum analysis components are not sorted according to their indices. This is unlike the empirical mode decomposition that the frequency bands of the intrinsic mode functions are sorted according to their indices. Because of this reason, the bandpass filtering is required to obtain the signal bands of the electroencephalogram.

### 2.2. Review on the Low-Rank Decomposition [18,19,20,21,22]

Let an observed signal y(n) be a combination of two components, namely the signal component s(n) and the noise component e(n). That is, y(n)=s(n)+e(n). Define the trajectory matrix of y(n), s(n) and e(n) as Y, S and E, respectively. Then, we have Y=S+E. Here, it is assumed that S only consists of few singular values. This implies that rank(S) is small. Besides, the total number of the nonzero elements in E is also assumed to be small. Denote the $L_{0}$ norm of a matrix Z as ${\left\|Z\right\|}_{0}$. Here, the $L_{0}$ norm of a matrix refers to the total number of the nonzero elements in that matrix. This implies that ${\left\|E\right\|}_{0}$ is small. As a result, the low-rank decomposition problem can be formulated as an optimization problem with the objective function being the weighted sum of rank(S) and ${\left\|E\right\|}_{0}$ subject to Y=S+E. Let the weight be γ. Then, we have:(5a)$$\underset{\left(S,E\right)}{\min\;}rank\left(S\right)+\gamma {\left\|E\right\|}_{0},$$
(5b)subject to Y=S+E.

However, as rank(S) is equal to the total number of the singular values of S, this operator is non-polynomial hard and nonconvex. To address this difficulty, rank(S) is approximated by the absolute sum of the singular values of S. Denote the nuclear norm of a matrix Z as ${\left\|Z\right\|}_{*}$. Here, the nuclear norm of a matrix refers to the absolute sum of the singular values of that matrix. Therefore, ${\left\|S\right\|}_{*}$ is minimized instead. Similarly, as ${\left\|E\right\|}_{0}$ refers to the total number of the nonzero elements in E, this operator is also non-polynomial hard and nonconvex. To address this difficulty, ${\left\|E\right\|}_{0}$ is approximated by the absolute sum of the elements in E. Denote the $L_{1}$ norm of a matrix Z as ${\left\|Z\right\|}_{1}$. Here, the $L_{1}$ norm of a matrix refers to the absolute sum of the nonzero elements in that matrix. Therefore, ${\left\|E\right\|}_{1}$ is also minimized instead. As a result, we have the following approximated problem:(6a)$$\underset{\left(S,E\right)}{\min\;}{\left\|S\right\|}_{*}+\gamma {\left\|E\right\|}_{1},$$
(6b)subject to Y=S+E.

## 3. Proposed Spike Suppression Method

Figure 1 shows the flowchart of the proposed method. Since most of the information of the electroencephalogram is found in the δ wave band (0–4 Hz), the θ wave band (4–8 Hz), the α wave band (8–12 Hz), the β wave band (12–39 Hz) and the γ wave band (30–44 Hz), these frequency bands are the most important frequency bands for performing the electroencephalograph analysis [1,2,3]. To suppress the noise outside these signal bands, a simple lowpass filtering is applied to the electroencephalogram. To implement this filtering operation, the simplest approach is to remove the discrete Fourier transform coefficients of the electroencephalogram outside these signal bands. As there is much existing hardware for the efficient implementation of the discrete Fourier transform [23,24,25], this filtering module is with very low cost. Here, the frequency response of the ideal lowpass filter is:(7)$$H\left(\omega \right)=\{\begin{array}{cc} 1, & \left|\omega \right|\leq \omega_{c}\\ 0, & \left|\omega \right|>\omega_{c} \end{array},$$
where $\omega_{c}$ is the cutoff frequency of the filter. Let x(t) be the original signal and X(ω) be the corresponding continuous time Fourier transform. Then, the filtered signal y(t) can be expressed as:(8)$$y(t)=\frac{1}{2\pi }\int_{-\omega_{c}}^{\omega_{c}}X(\omega )H\left(\omega \right)e^{j\omega t}d\omega .$$

To suppress the spikes in the filtered electroencephalogram, the filtered electroencephalogram is required to decompose into various components and appropriate processing is applied to these components. Since the magnitudes of the spikes are large [4,5,6] and the singular spectrum analysis components are expressed as the magnitudes of the eigenvalues of the trajectory matrix [15,16,17], the singular spectrum analysis is an appropriate tool to decompose the filtered electroencephalogram into various components for suppressing the spikes.

By performing the singular spectrum analysis on the filtered electroencephalogram, various singular spectrum analysis components are obtained. First, the singular spectrum analysis components of the filtered electroencephalogram are sorted in the descending order of the magnitudes of the corresponding eigenvalues. Since the magnitudes of the spikes are large [4,5,6], the singular spectrum analysis components corresponding to the small eigenvalues do not contain the spikes. Therefore, the singular spectrum analysis components corresponding to the small eigenvalues are summed up together to obtain an approximated despiked electroencephalogram. This is unlike the conventional singular spectrum analysis based denoising methods [15,16,17] that the singular spectrum analysis components corresponding to small eigenvalues are discarded. The obtained approximated despiked electroencephalogram is called the first scale of the electroencephalogram. Mathematically, let the sorted singular spectrum analysis components be ${\overset{⌣}{x}}_{m}$ for $m=1,\cdots ,\tilde{M}$. Let the first index of the singular spectrum analysis component to be summed together be $m^{*}$. Let $\kappa_{0}$ be the first scale of the electroencephalogram. Then, we have $\kappa_{0}=\sum_{m=m^{*}}^{\tilde{M}}{\overset{⌣}{x}}_{m}$. Here, it is required to determine $m^{*}$ via a thresholding method. That is, the singular spectrum analysis components are categorized into two groups, namely the group containing the singular spectrum analysis components corresponding to the large eigenvalues and the group containing the singular spectrum analysis components corresponding to the small eigenvalues, via a thresholding method. Since the singular spectrum analysis components are sequentially added together starting from the last singular spectrum analysis component, if the variance of the summed singular spectrum analysis component under the unit energy normalization is larger than a threshold value, then the summation is terminated. The singular spectrum analysis components involved in the summed singular spectrum analysis component are assigned to the group corresponding to the singular spectrum analysis components corresponding to the small eigenvalues. On the other hand, the remaining singular spectrum analysis components are assigned to the group corresponding to the singular spectrum analysis components corresponding to the large eigenvalues.

However, the sum of the singular spectrum analysis components corresponding to the large eigenvalues usually contains useful information of the electroencephalogram. Hence, further processing is required to apply to the sum of these singular spectrum analysis components corresponding to the large eigenvalues. As the low-rank decomposition can effectively decompose a signal into the low-rank component and the sparse component as well as the low-rank component does not contain the spikes [18,19,20], the low-rank decomposition is used to decompose the sum of the singular spectrum analysis components corresponding to the large eigenvalues and the low-rank component is added to the current scale of the electroencephalogram to obtain the next scale of the electroencephalogram. Mathematically, let the low-rank component and the sparse component of the sum of the singular spectrum analysis components corresponding to the large eigenvalues be ${\overset{⌣}{e}}_{0}$ and ${\overset{⌣}{s}}_{0}$, respectively. That is, $\sum_{m=1}^{m^{*}-1}{\overset{⌣}{x}}_{m}={\overset{⌣}{e}}_{0}+{\overset{⌣}{s}}_{0}$. Let $\kappa_{1}$ be the second scale of the electroencephalogram. Then, we have $\kappa_{1}=\kappa_{0}+{\overset{⌣}{e}}_{0}$.

In order to further dig out the useful information from the sparse component, the variance of the current scale of the electroencephalogram under the unit energy normalization is computed. If this variance is smaller than another threshold value, then the above procedures are iterated on the sparse component. Otherwise, the iterative algorithm is terminated, and the final scale of the electroencephalogram is taken as the despiked electroencephalogram.

It is worth noting that the proposed algorithm enjoys the multi-resolution property like the conventional wavelet-based multi-resolution property where the wavelet coefficients are added to the current scale of the signal and the original signal is reconstructed scale by scale progressively. Here, the singular spectrum analysis components corresponding to the small eigenvalues and the low-rank components are added to the current scales of the electroencephalogram and the despiked electroencephalogram is reconstructed scale by scale progressively.

## 4. Computer Numerical Simulation Results

The proposed algorithm is implemented under the MATLAB 2018a environment and the computer numerical simulations are conducted using the Core i7-6700 3.41 GHz CPU with 8 GB RAM. In this paper, the electroencephalograms are acquired by a MUSE2 headband. The MUSE2 headband acquires the electroencephalograms at four different locations of the head. Two are at the forehead and two are behind the ear. In the following computer numerical simulations, the electroencephalograms are taken from the left forehead. The sampling rate of the electroencephalograms is 100 Hz and the acquired electroencephalograms are downloaded via the mobile application unit built in the MUSE2 headband. It is worth noting that the acquired electroencephalograms have significant changes in the amplitudes in the short periods of times at the locations where the spikes occur, and the locations of the spikes are random.

### 4.1. Computer Numerical Simulation Results

Figure 2a,b show a realization of a section of an acquired electroencephalogram in both the time domain and the frequency domain, respectively. That is, Figure 2a shows a section of the entire electroencephalogram in the time domain. Figure 2b shows the same section of the entire electroencephalogram in the frequency domain. It can be seen from Figure 2a that there are some spikes corrupted in the acquired electroencephalogram. Also, it can be seen from Figure 2b that the acquired electroencephalogram is a wide spectrum signal. Figure 3 shows the locations of the spikes that need to be suppressed in the time domain. Figure 3 also shows that there are 11 spikes in the acquired electroencephalogram and the acquired electroencephalogram has significant changes in the amplitudes in the short periods of times at the locations where the spikes occur. Figure 4a shows the same section of the entire electroencephalogram after applying the ideal lowpass filtering in the time domain. Figure 4b shows the same section of the entire electroencephalogram after applying the ideal lowpass filtering in the frequency domain. Here, since most of the information of the electroencephalogram is found in the 0–50 Hz frequency band, $\omega_{c}=50\;\mathrm{Hz}$ is chosen. It can be seen from Figure 4b that the noise outside the signal bands is removed after performing the ideal lowpass filtering. Although it can be seen from Figure 4a that these 11 spikes are still there, the background noise is significantly suppressed.

Figure 5 shows the singular spectrum analysis components obtained by performing the singular spectrum analysis on the filtered electroencephalogram based on the discussion presented in Section 2.1. It is worth noting that a too large value of L requires a heavy computational power while a too small value of L does not contain enough of the singular spectrum analysis components to be selected for performing the further processing. Therefore, L=10 is chosen in the following computer numerical simulation results. This value is a good tradeoff between the above two factors. Also, as it is 0.5% of the length of the section of an acquired electroencephalogram which is less than half of the length of the section of an acquired electroencephalogram, it satisfies the property of the singular spectrum analysis. Figure 6 shows the sums of the singular spectrum analysis components starting from the last singular spectrum analysis component. That is, the singular spectrum analysis component shown in Figure 6a is the same as that shown in Figure 5j. The singular spectrum analysis component shown in Figure 6b is the sum of the singular spectrum analysis components shown in Figure 5i,j. The singular spectrum analysis component shown in Figure 6c is the sum of the singular spectrum analysis components shown in Figure 5h–j. The rest of the singular spectrum analysis components shown in Figure 6 are the sums of the singular spectrum analysis components shown in corresponding subfigures in Figure 5. Finally, the singular spectrum analysis component shown in Figure 6j is the sum of the singular spectrum analysis components in all the subfigures in Figure 5. In fact, Figure 6j is the filtered electroencephalogram. Table 1 lists the values of the variances of the sums of the singular spectrum analysis components under the unit energy normalization. When comparing Figure 6j with other figures in Figure 6, it can be seen that there are spikes in the sums of the singular spectrum analysis components if the summations added from the last spectrum analysis component to the singular spectrum analysis components corresponding to the indices larger than or equal to four. Therefore, the threshold value defined on the variances of the sums of the singular spectrum analysis components under the unit energy normalization is set at $10^{-7}$ such that only the sum of the fifth singular spectrum analysis component to the last singular spectrum analysis component is used to generate the first scale of the electroencephalogram.

Figure 7a,b show the first scale of the electroencephalogram and the residue of the electroencephalogram, respectively. It can be seen from Figure 7a that there is no spike in the first scale of the electroencephalogram, while all the spikes are found in the residue of the electroencephalogram. Figure 8a,b show the low-rank component and the sparse component after performing the low-rank decomposition on the residue of the electroencephalogram. Here, since both the sparse component and the low-rank component have the same effects on the results, let γ=0.5. Similarly, it can be seen from Figure 8a that there is no spike in the low-rank component, while all the spikes are found in the sparse component. By adding the low-rank component to the first scale of the electroencephalogram, we obtain the second scale of the electroencephalogram which is shown in Figure 9.

To further dig out the useful information of the sparse component, the singular spectrum analysis is applied to the sparse component to obtain a new set of the singular spectrum analysis components. The new set of the singular spectrum analysis components corresponding to the small eigenvalues are added to the current scale of the electroencephalogram to obtain the next scale of the electroencephalogram. Also, the low-rank decomposition is applied to the new residue of the electroencephalogram. The new low-rank components are also added to the current scale of the electroencephalogram to obtain the next scale of the electroencephalogram. Here, these scales of the electroencephalogram do not contain the spikes, but small amounts of useful information are lost. In order to recover the discarded useful information, the above procedures are repeated on the sparse component. From here, it is worth noting that the spikes are not gradually suppressed in each iteration. On the other hand, the useful information is added to the scales of the electroencephalogram. Figure 10 and Figure 11 show the 2*n*th scales of the electroencephalogram in the time domain and the frequency domain, respectively. It can be seen from Figure 10c that there is no spike after performing three iterations. On the other hand, it can be seen from Figure 10d that there are spikes after performing four iterations. Compared with Figure 11b,c, it can be seen that the electroencephalogram after performing two iterations lost some low-frequency information. Compared with Figure 11c,d, the low-frequency information of the electroencephalogram after performing four iterations does not increase significantly. Table 2 shows the variances of the 2*n*th scales of the electroencephalogram under the unit energy normalization. Here, the threshold value defined on the variances of the 2*n*th scales of the electroencephalogram under the unit energy normalization is set at 10 such that only three iterations are performed.

Figure 12 shows the despiked electroencephalograms based on the same set of the threshold values for the new realizations of an electroencephalogram. It can be seen from Figure 12 that our proposed method still achieves the good despiked performances. This is because the same type of electroencephalograms is acquired by the same device. Therefore, the acquired electroencephalograms have consistent characteristics. After the threshold values are predefined, it is not required to change the predefined threshold values for the new realization of the electroencephalogram.

### 4.2. Comparisons to the Existing Methods

In order to demonstrate the effectiveness of our proposed method, our proposed method is compared to the following three states of the art methods. They are the empirical mode decomposition based denoising method [13,14], the singular spectrum analysis based denoising method [15,16,17] and the low-rank decomposition based denoising method [18,19,20].

Since the spikes contain the high-frequency contents of the electroencephalogram, the empirical mode decomposition based denoising method reconstructs the despiked electroencephalogram using the intrinsic mode functions corresponding to the low-frequency components [13,14]. Here, the intrinsic mode functions are obtained by applying the empirical mode decomposition on the original electroencephalogram. It is worth noting that if too many intrinsic mode functions are used for reconstructing the despiked electroencephalogram, then the despiked electroencephalogram will still contain the spikes. On the other hand, if very few intrinsic mode functions are used for reconstructing the despiked electroencephalogram, then the despiked electroencephalogram will be very smooth and some useful information will be lost. To determine which intrinsic mode functions are used for reconstructing the despiked electroencephalogram, Figure 13 and Figure 14 show the despiked electroencephalogram reconstructed using the last five intrinsic mode functions, the last seven intrinsic mode functions and the last nine intrinsic mode functions shown in the time domain and in the frequency domain, respectively. On the other hand, Figure 15 shows the despiked electroencephalogram in the time domain obtained using our proposed approach. Our proposed approach can significantly suppress the spikes while most of the information is retained in the despiked electroencephalogram. Figure 16a shows the despiked electroencephalogram in the frequency domain obtained using our proposed approach. The main lobe of the despiked electroencephalogram obtained using our proposed approach is mainly localized within 0–10 Hz while there is a large second lobe component localized between 10 Hz and 50 Hz. From here, we can reconstruct the despiked electroencephalogram by summing up the intrinsic mode functions from the one with the lowest frequency component to the one such that the cutoff frequency of the summed intrinsic mode functions is located at 10 Hz. From Figure 14, we can see that the cutoff frequency of the despiked electroencephalogram reconstructed using the last five intrinsic mode functions is around 5 Hz, which is too low, resulting in a large amount of information being lost. On the other hand, the despiked electroencephalogram reconstructed using the last nine intrinsic mode functions contain too many high-frequency noises. Whereas, the despiked electroencephalogram reconstructed using the last seven intrinsic mode functions achieves the best tradeoff results. Therefore, the despiked electroencephalograms are reconstructed using the last seven intrinsic mode functions and it is shown in Figure 14. However, it can be seen from Figure 15 that the spikes are still found in the despiked electroencephalogram obtained based on the conventional empirical mode decomposition-based method. Compared to our obtained despiked electroencephalogram, it can be concluded that our proposed method outperforms the empirical mode decomposition-based method in terms of suppressing the spikes.

The reconstructed electroencephalograms obtained based on our proposed method and the conventional singular spectrum analysis-based method are shown in Figure 17. Here, the conventional singular spectrum analysis-based method reconstructs the electroencephalogram by summing up the fifth singular spectrum analysis component to the last singular spectrum analysis component of the filtered electroencephalogram. On the other hand, as our proposed method also sums up the fifth singular spectrum analysis component to the last singular spectrum analysis component of the filtered electroencephalogram to generate the first scale of the electroencephalogram, the reconstructed electroencephalogram obtained based on our proposed method is similar to that based on the conventional singular spectrum analysis based method [15,16,17]. However, it can be seen from Figure 17 that the reconstructed electroencephalogram obtained based on the conventional singular spectrum analysis-based method is over-smoothen. This is because the information in the first four singular spectrum analysis components is lost. Compared with our reconstructed despiked electroencephalogram, it can be concluded that our proposed method outperforms the conventional singular spectrum analysis-based method in terms of retaining the characteristics of the electroencephalogram.

Finally, the low-rank decomposition-based method [18,19,20,21,22] is also compared. Here, the low-rank component obtained by applying the low-rank decomposition to the filtered electroencephalogram is taken as the despiked electroencephalogram. This is because the spikes are found in the sparse component. The electroencephalograms obtained based on our proposed method and the conventional low-rank decomposition-based method are shown in Figure 16. Similar to the conventional singular spectrum analysis approach [15,16,17], it can be understood from Figure 18 that the reconstructed electroencephalogram based on the low-rank decomposition-based method is over-smoothen. This is because the information in the sparse component is lost. Compared with our reconstructed despiked electroencephalogram, it can be concluded that our proposed method outperforms the low-rank decomposition-based method in terms of retaining the characteristics of the electroencephalogram.

It can be seen in Figure 3 that there are 11 spikes in the electroencephalogram. Here, the objective is to suppress the magnitudes of these 11 spikes. However, it is worth noting that the waveforms used in the computer numerical simulations are the realizations of a practical electroencephalogram. There is no external noise added to the electroencephalogram. Since there is no clean electroencephalogram for performing the comparison, neither the noise level nor the signal to noise ratio cannot be measured. Therefore, it is difficult to have a comparison of quantitative performance. To address this issue, Table 3 shows the differences in the magnitudes of the spikes at these 11 points before and after applying the above four despiked methods. It can be understood from Table 3 that the empirical mode decomposition-based method does not effectively suppress the magnitudes of the spikes. Figure 16 shows the reconstructed electroencephalograms in the frequency domain obtained based on the above four methods. It can be seen in Figure 16 that our proposed method does not lose too much information in the low-frequency content of the electroencephalogram. On the other hand, the singular spectrum analysis-based method loses the information in the low-frequency content of the electroencephalogram, while the low-rank decomposition-based method loses the information in the high-frequency content of the electroencephalogram. Hence, it can be concluded that our proposed method outperforms the other three methods in terms of suppressing the spikes as well as retaining the information of the electroencephalogram.

To further demonstrate the outperformance of our proposed method, a synthetic signal is illustrated. The clean signal $s^{*}\left(n\right)\in ℜ\left[n\right]$ is given by
(9)$$s^{*}\left(n\right)=10\sin2\pi f_{1}n+4\sin(2\pi f_{2}n+\sin2\pi f_{3}n)+3\sin(2\pi f_{3}n+\sin2\pi f_{4}n).$$

Here, $f_{1}=1$, $f_{2}=10$, $f_{3}=30$ and $f_{4}=50$. Also, the spike signal $\eta^{*}\left(n\right)\in ℜ\left[n\right]$ composed of the convolution of two square waves with the amplitude and the position of the spike in the spike signal being random is corrupted to $s^{*}\left(n\right)$. That is, the synthetic signal is $s\left(n\right)=s^{*}\left(n\right)+\eta^{*}\left(n\right)$ and it is shown in Figure 19c. To perform the quantitative evaluation, the signal to noise ratio criterion defined as $10\log_{10}\frac{\sum_{n=1}^{N}{\left(s\left(n\right)\right)}^{2}}{\sum_{n=1}^{N}{\left(\eta^{*}\left(n\right)\right)}^{2}}$ is evaluated. By changing the range of the amplitude and the distribution of the position of the spike, the synthesized signals with different signal to noise ratios are obtained. Table 4 shows the signal to noise ratios of the despiked signals obtained by the above four despiked methods. For our proposed method, L=10 and γ=0.5 are chosen. Moreover, the threshold value for selecting the singular spectrum analysis components is set in such a way that the first four singular spectrum analysis components are chosen. Furthermore, the threshold value for terminating the algorithm is set in such a way that the eighth scale of the synthetic signal is the final despiked signal. For the singular spectrum analysis based denoising method, the sum of the first fourth singular spectrum analysis components is chosen as the despiked signal. For the empirical mode decomposition based denoising method, the sum of the last seven intrinsic mode functions is chosen as the despiked signal. For the low-rank decomposition based denoising method, γ=0.5 is chosen. All of these parameters are chosen as the same as that before. As shown in Table 4, our proposed method outperforms the other three denoising methods in terms of suppressing the spikes. This is because the signal to noise ratio achieved by our proposed method is 20 dB larger than the other three denoising methods.

## 5. Conclusions

This paper has proposed an iterative joint singular spectrum analysis and low-rank decomposition method for suppressing the spikes in an electroencephalogram. Since the spikes are with large magnitudes and they are sparse, both the singular spectrum analysis-based denoising method and the low-rank decomposition-based denoising method can effectively suppress the spikes. The computer numerical simulation results show that our proposed method outperforms the states of the art methods in terms of suppressing the spikes and retaining the information of the electroencephalogram.

## Figures and Tables

**Figure 1 sensors-20-00341-f001:**
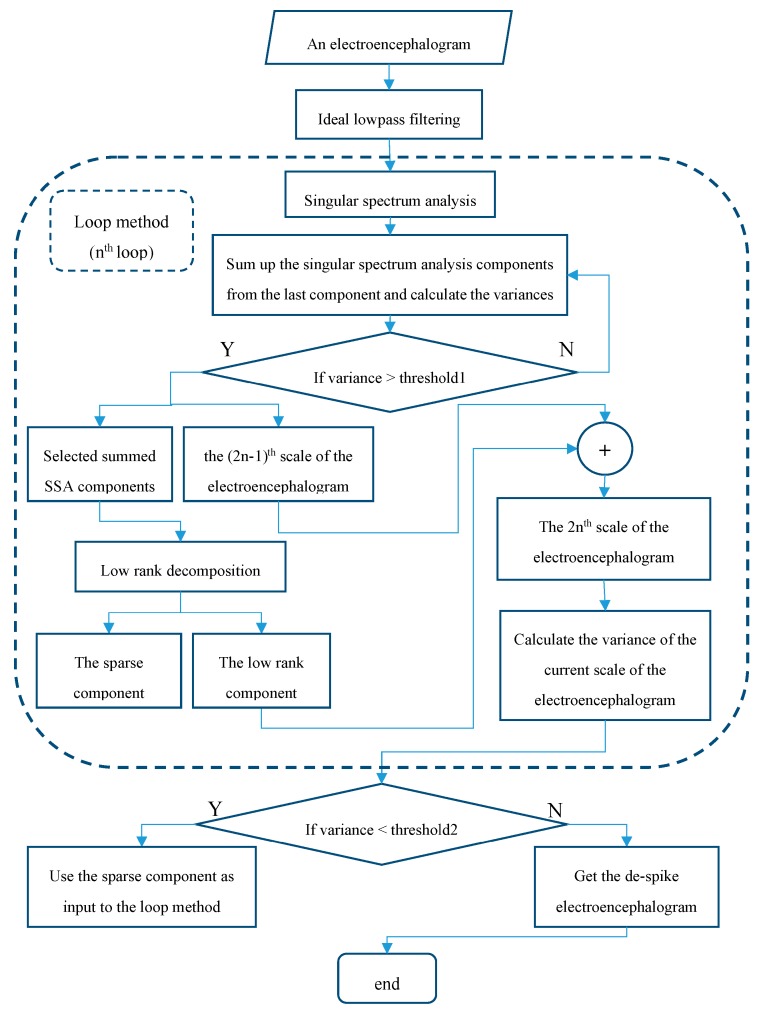
The flowchart of the proposed method.

**Figure 2 sensors-20-00341-f002:**
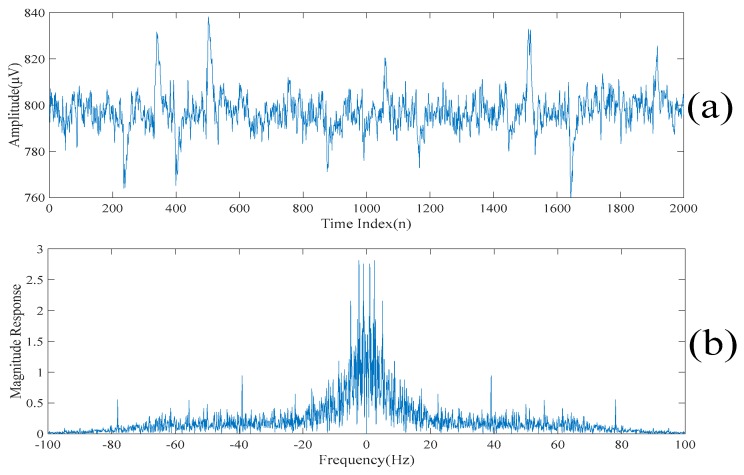
(**a**) A realization of a section of an acquired electroencephalogram in the time domain. (**b**) A realization of a section of an acquired electroencephalogram in the frequency domain.

**Figure 3 sensors-20-00341-f003:**
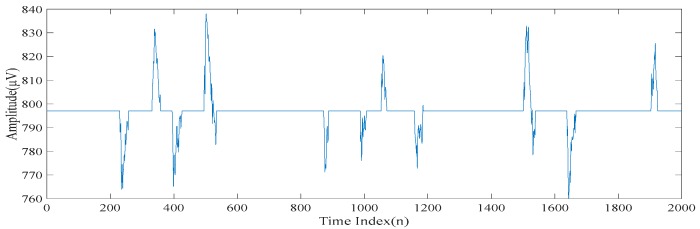
The locations of the spikes that need to be suppressed.

**Figure 4 sensors-20-00341-f004:**
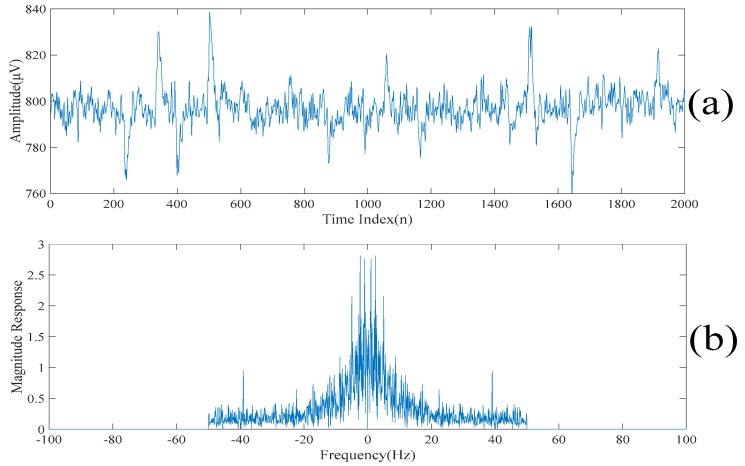
(**a**) The impulsive response of the filtered electroencephalogram. (**b**) The magnitude response of the filtered electroencephalogram.

**Figure 5 sensors-20-00341-f005:**
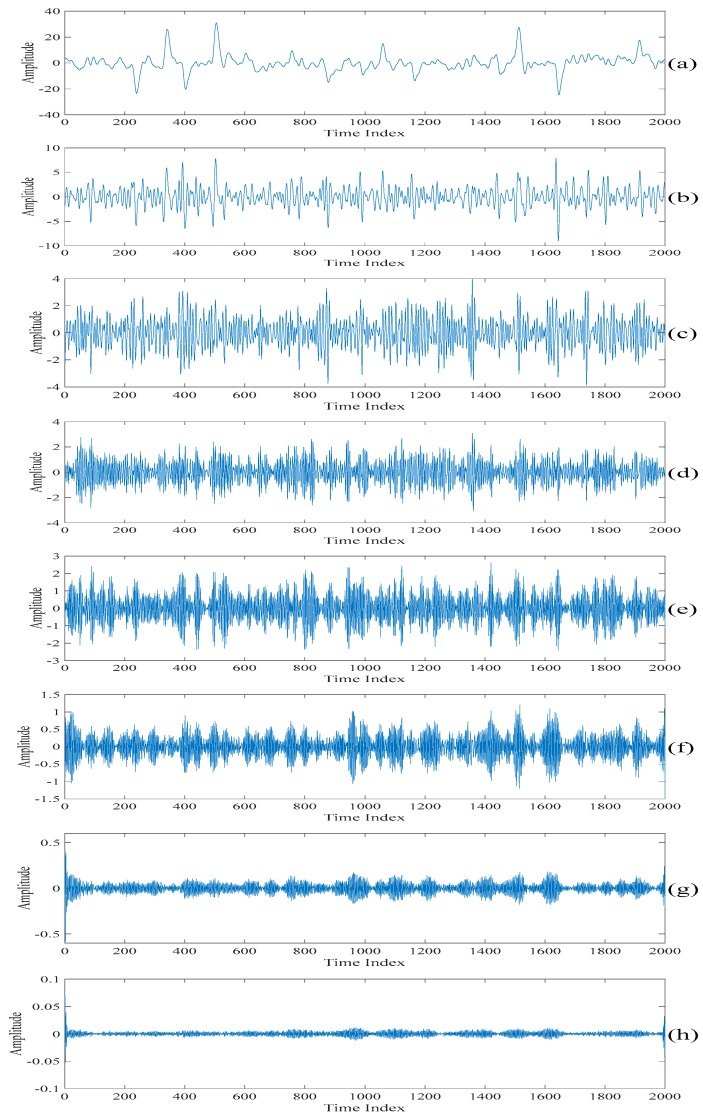
The singular spectrum analysis components obtained by performing the singular spectrum analysis on the filtered electroencephalogram. (**a**) The first singular spectrum analysis component. (**b**) The second singular spectrum analysis component. (**c**) The third singular spectrum analysis component. (**d**) The fourth singular spectrum analysis component. (**e**) The fifth singular spectrum analysis component. (**f**) The sixth singular spectrum analysis component. (**g**) The seventh singular spectrum analysis component. (**h**) The eighth singular spectrum analysis component. (**i**) The ninth singular spectrum analysis component. (**j**) The tenth singular spectrum analysis component.

**Figure 6 sensors-20-00341-f006:**
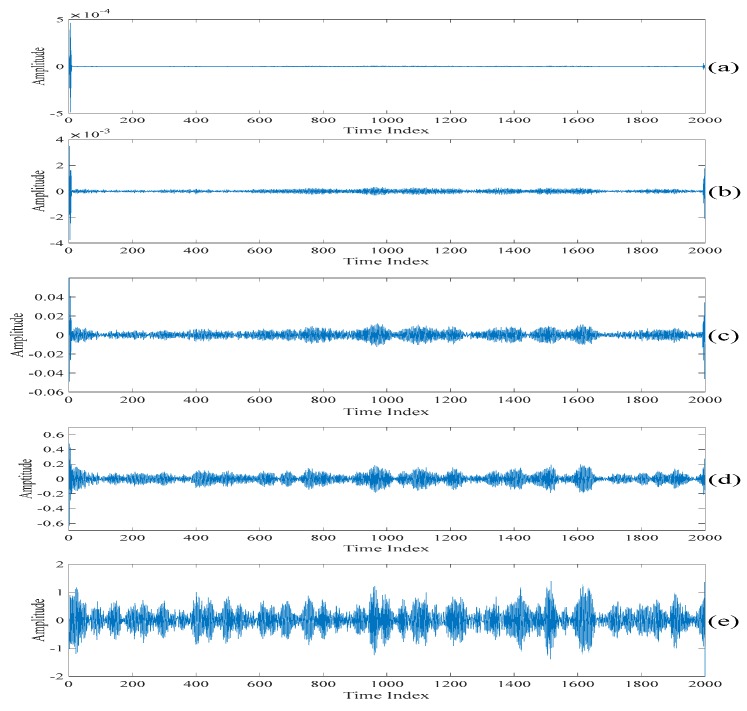
The sums of the singular spectrum analysis components starting from the last singular spectrum analysis component. (**a**) The last singular spectrum analysis component. (**b**) The sum of the last two singular spectrum analysis components. (**c**) The sum of the last three singular spectrum analysis components. (**d**) The sum of the last four singular spectrum analysis components. (**e**) The sum of the last five singular spectrum analysis components. (**f**) The sum of the last six singular spectrum analysis components. (**g**) The sum of the last seven singular spectrum analysis components. (**h**) The sum of the last eight singular spectrum analysis components. (**i**) The sum of the last nine singular spectrum analysis components. (**j**) The sum of all the singular spectrum analysis components.

**Figure 7 sensors-20-00341-f007:**
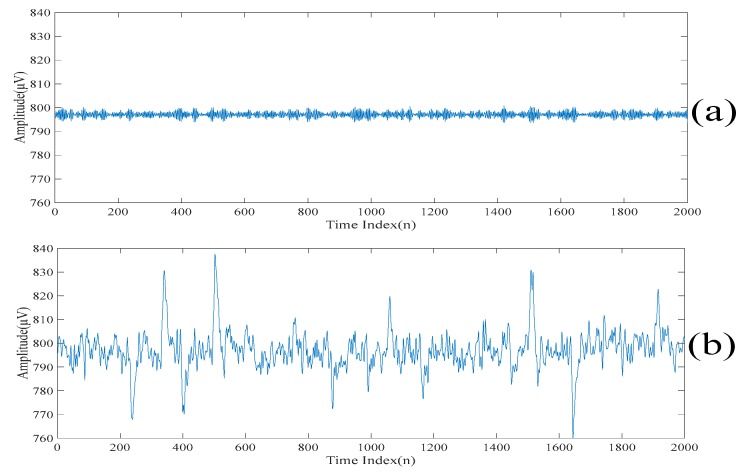
(**a**) The first scale of the electroencephalogram. (**b**) The residue of the electroencephalogram.

**Figure 8 sensors-20-00341-f008:**
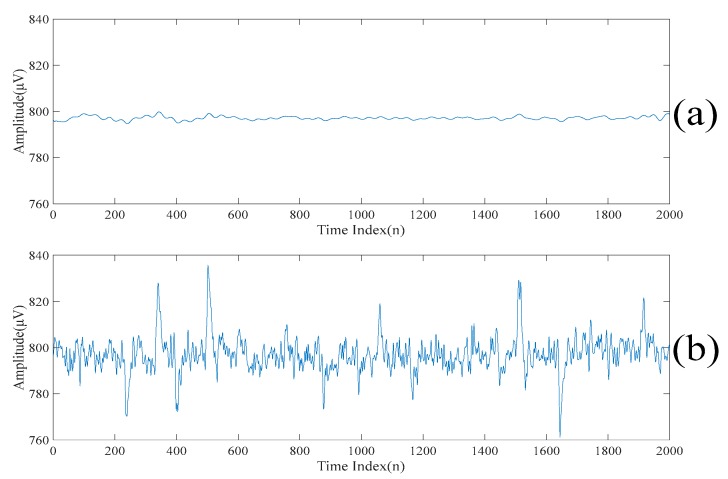
(**a**) The low-rank component after performing the low-rank decomposition on the residue of the electroencephalogram. (**b**) The sparse component after performing the low-rank decomposition on the residue of the electroencephalogram.

**Figure 9 sensors-20-00341-f009:**
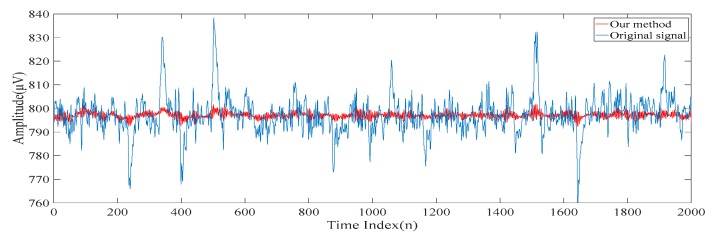
The second scale of the electroencephalogram.

**Figure 10 sensors-20-00341-f010:**
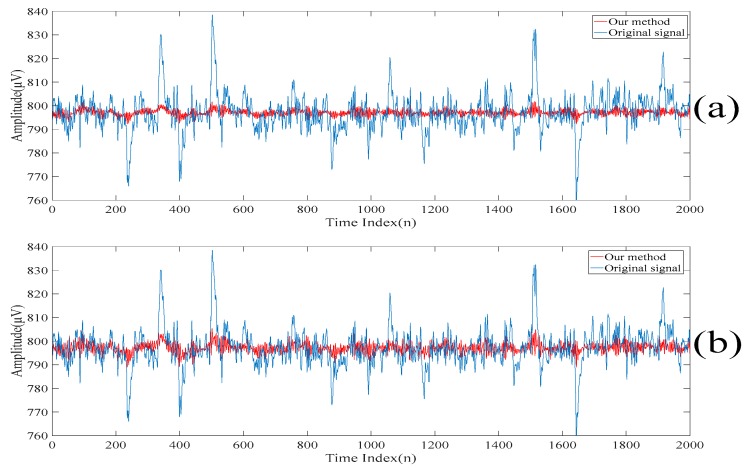
(**a**) The second scale of the electroencephalogram in the time domain. (**b**) The fourth scale of the electroencephalogram in the time domain. (**c**) The sixth scale of the electroencephalogram in the time domain. (**d**) The eighth scale of the electroencephalogram in the time domain. (**e**) The tenth scale of the electroencephalogram in the time domain.

**Figure 11 sensors-20-00341-f011:**
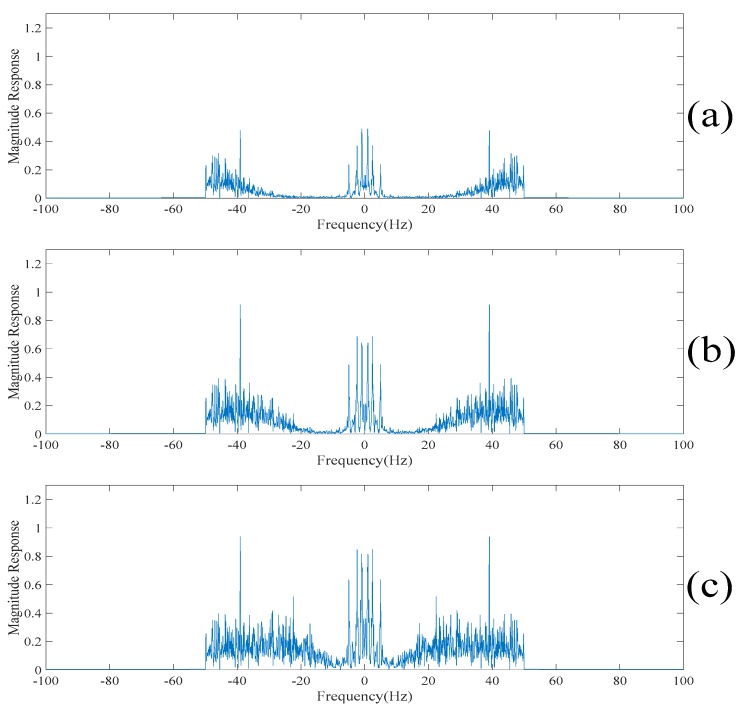
(**a**) The second scale of the electroencephalogram in the frequency domain. (**b**) The fourth scale of the electroencephalogram in the frequency domain. (**c**) The sixth scale of the electroencephalogram in the frequency domain. (**d**) The eighth scale of the electroencephalogram in the frequency domain. (**e**) The tenth scale of the electroencephalogram in the frequency domain.

**Figure 12 sensors-20-00341-f012:**
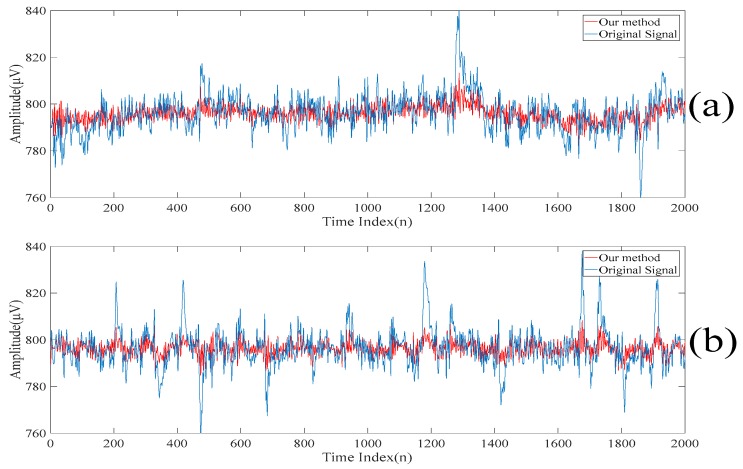
(**a**) The despiked electroencephalogram based on the same set of the threshold values for a new realization of an electroencephalogram. (**b**) The despiked electroencephalogram based on the same set of the threshold values for a new realization of an electroencephalogram.

**Figure 13 sensors-20-00341-f013:**
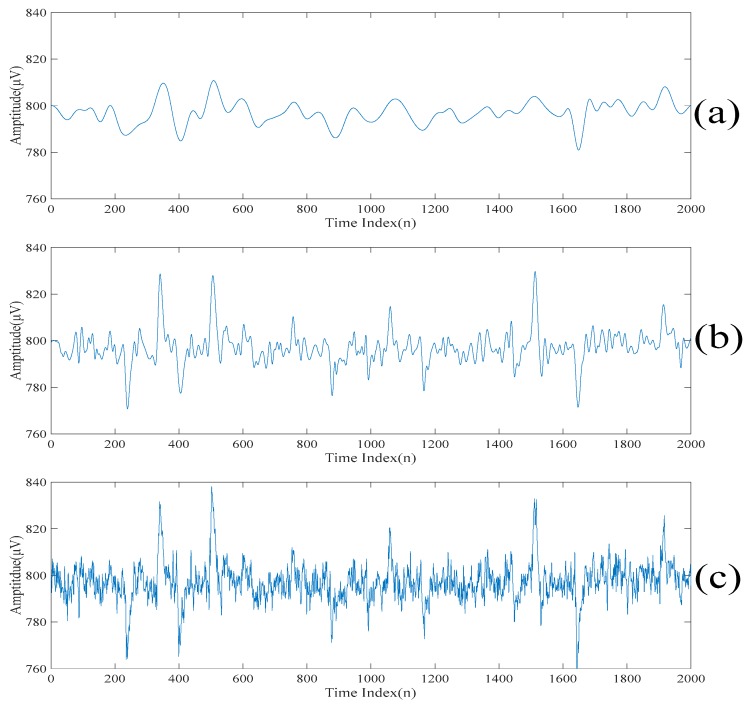
(**a**) The despiked electroencephalogram in the time domain reconstructed using the last five intrinsic mode functions. (**b**) The despiked electroencephalogram in the time domain reconstructed using the last seven intrinsic mode functions. (**c**) The despiked electroencephalogram in the time domain reconstructed using the last nine intrinsic mode functions.

**Figure 14 sensors-20-00341-f014:**
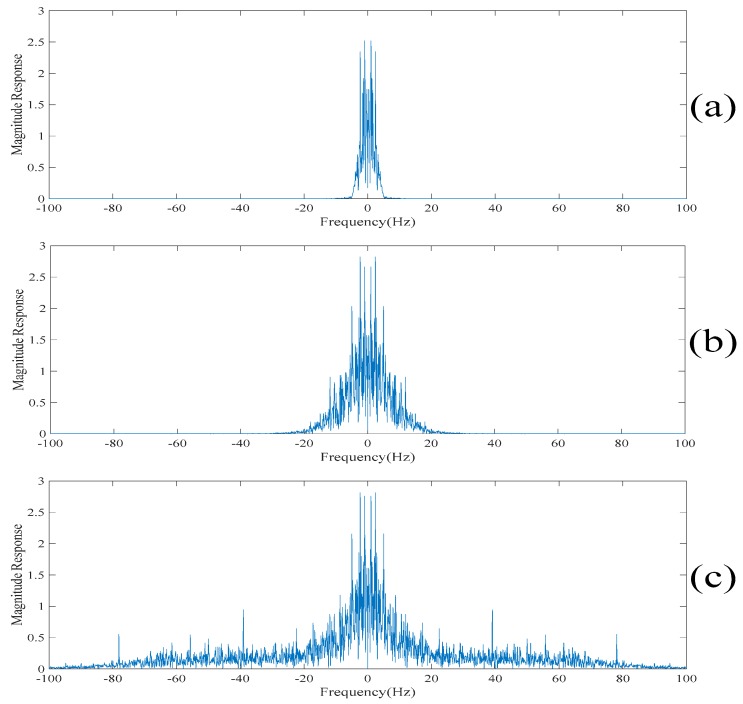
(**a**) The despiked electroencephalogram in the frequency domain reconstructed using the last five intrinsic mode functions. (**b**) The despiked electroencephalogram in the frequency domain reconstructed using the last seven intrinsic mode functions. (**c**) The despiked electroencephalogram in the frequency domain reconstructed using the last nine intrinsic mode functions.

**Figure 15 sensors-20-00341-f015:**
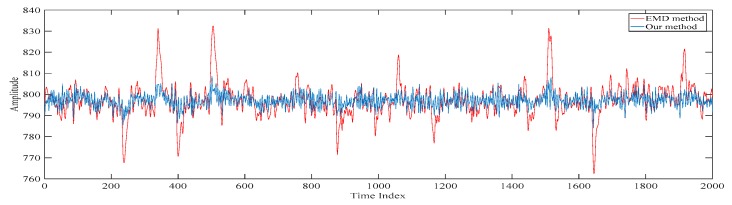
The despiked electroencephalograms obtained based on our proposed method and the conventional empirical mode decomposition-based method.

**Figure 16 sensors-20-00341-f016:**
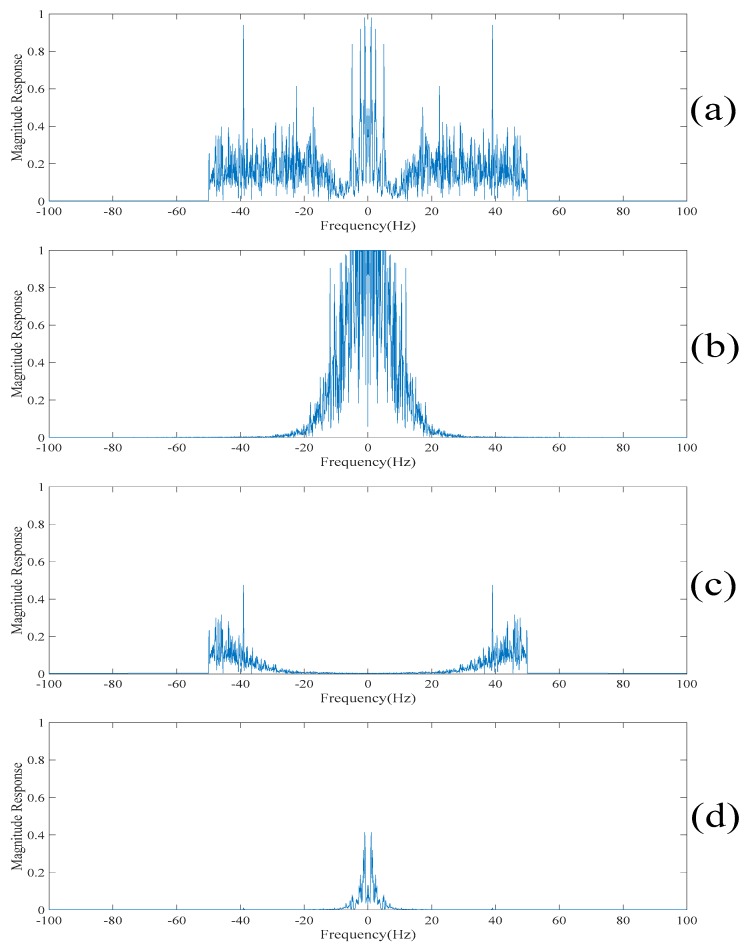
(**a**) The reconstructed electroencephalogram in the frequency domain obtained based on our proposed method. (**b**) The reconstructed electroencephalogram in the frequency domain obtained based on the conventional empirical mode decomposition-based method. (**c**) The reconstructed electroencephalogram in the frequency domain obtained based on the conventional singular spectrum analysis-based method. (**d**) The reconstructed electroencephalograms in the frequency domain obtained based on the conventional low-rank decomposition-based method.

**Figure 17 sensors-20-00341-f017:**
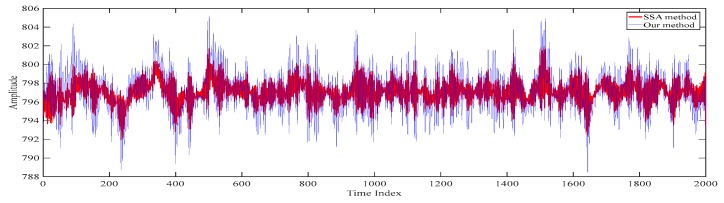
The reconstructed electroencephalograms obtained based on our proposed method and the conventional singular spectrum analysis-based method.

**Figure 18 sensors-20-00341-f018:**
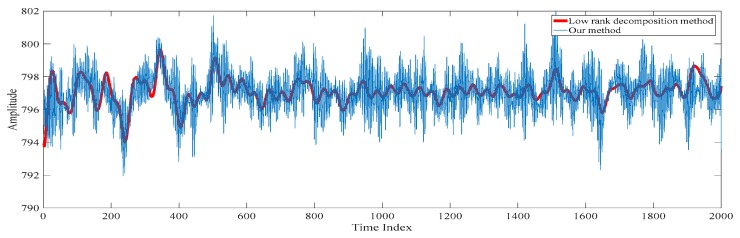
The electroencephalograms obtained based on our proposed method and the conventional low-rank decomposition-based method.

**Figure 19 sensors-20-00341-f019:**
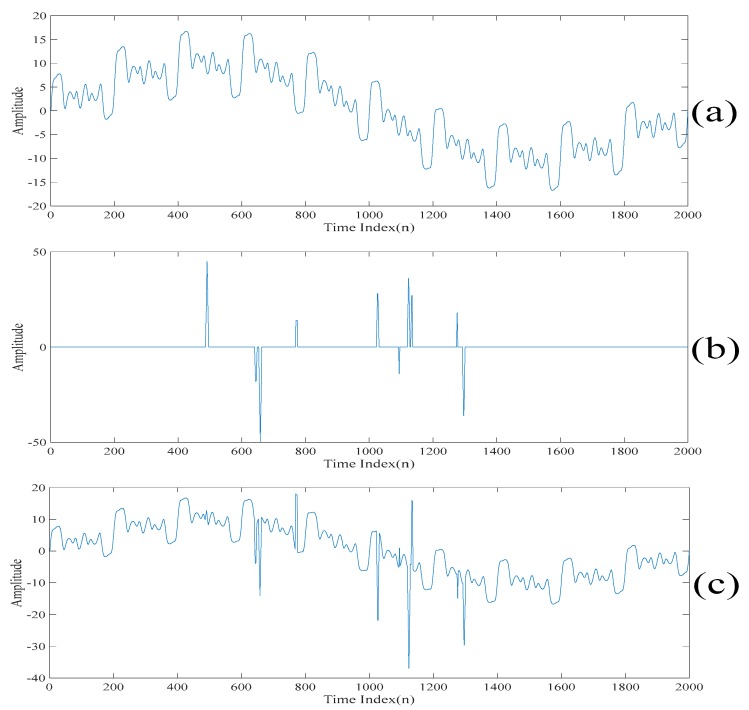
(**a**) The clean signal $s^{*}\left(n\right)$. (**b**) The spike signal $\eta^{*}\left(n\right)$. (**c**) The synthetic signal s(n).

**Table 1 sensors-20-00341-t001:** The variances of the sums of the singular spectrum analysis components under the unit energy normalization.

$\mathrm{The}\;\mathrm{Sum}\;\mathrm{from}\;\mathrm{the}\;k^{\mathrm{th}}\;\mathrm{Sin}\mathrm{gular}\;\mathrm{Spectrum}\;\mathrm{Analysis}\;\mathrm{Component}\;\mathrm{to}\;\mathrm{the}\;\mathrm{Last}\;\mathrm{Sin}\mathrm{gular}\;\mathrm{Spectrum}\;\mathrm{Analysis}\;\mathrm{Component}$	Variance
k=1	3.93E-09
k=2	1.95E-08
k=3	4.12E-08
k=4	7.16E-08
k=5	1.91E-07
k=6	1.52E-06
k=7	5.54E-05
k=8	1.09E-02
k=9	6.71
k=10	625

**Table 2 sensors-20-00341-t002:** The variances of the 2*n*th scales of the electroencephalogram under the unit energy normalization.

Iteration Index	Variance
1	1.697772
2	5.202817
3	9.160693
4	11.97295
5	14.3795

**Table 3 sensors-20-00341-t003:** The differences on the magnitudes of the spikes at these 11 points before and after applying the above four despiked methods.

Spike	1	2	3	4	5	6	7	8	9	10	11
Our method	−18.669	21.035	−13.831	25.604	−13.955	−10.321	15.600	−12.295	22.358	−22.999	15.499
EMD	−1.209	0.457	−16.581	14.012	−3.305	−7.149	6.157	−6.468	3.277	−15.295	11.497
SSA	−28.519	33.465	−26.249	40.279	−24.582	−17.183	21.982	−20.571	32.205	−37.171	24.499
LRD	−25.175	29.609	−30.222	39.398	−22.269	−21.019	22.396	−23.211	31.228	−36.574	28.262

**Table 4 sensors-20-00341-t004:** The signal to noise ratios of the despiked signals obtained by the above four denoising methods.

Computer Numerical Simulation Index	Signal to Noise Ratio of the Original Signal	Signal to Noise Ratio of the Despiked Signal Obtained by the Empirical Mode Decomposition based Denoising Method	Signal to Noise Ratio of the Despiked Signal Obtained by the Singular Spectral Analysis based Denoising Method	Signal to Noise Ratio of the Despiked Signal Obtained by the Low-Rank Decomposition Based Denoising Method	Signal to Noise Ratio of the Despiked Signal Obtained by Our Proposed Method
1	51.158	25.950	51.572	13.371	**55.932**
2	40.705	23.291	40.948	13.097	**54.321**
3	30.386	20.222	30.513	12.769	**47.815**
4	20.085	20.464	20.649	12.531	**37.468**
5	10.905	−1.903	11.016	11.212	**26.661**
6	0.949	2.671	1.169	9.497	**18.986**
7	−1.068	−21.433	−0.698	9.391	**16.539**
8	−10.080	−31.536	−9.912	8.790	**12.203**
9	−20.386	−52.846	−20.212	**4.067**	2.909
10	−30.792	−38.598	−30.637	**5.701**	3.396

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
