# Peer review of "Suppressing the Spikes in Electroencephalogram via an Iterative Joint Singular Spectrum Analysis and Low-Rank Decomposition Approach"

_sensors, 2020, doi:10.3390/s20020341_

Round 1

Reviewer 1 Report

Please consult the attached .pdf file of the manuscript with sticky notes providing in detail my comments and corrections that need to be made.

Reviewer 2 Report

Title: Spike Removal of Electroencephalogram via an Iterative Joint Singular Spectrum Analysis and Low Rank Decomposition Approach
Authors: Zikang Tian, Bingo Wing-Kuen Ling, Xueling Zhou, Ringo Wai-Kit Lam and Kok-Lay Teo

This paper introduces an EEG spike removal technique through an iterative joint singular spectrum analysis and low rank decomposition approach. The paper addresses that both techniques effectively suppress spikes due to their larger magnitudes and sparsity.

The below suggestions must be applied to the paper:

Abstract:

The abstract has been written very weakly. It only introduces an abbreviated version of the method and does not discuss novelty, contribution or provides findings!

Proposed spike removal method:

My main concern here is how the threshold value 1 and 2 were chosen/set? A detailed information is required on this for

Section 4.1:

In lines 198-9 and 203, it feels necessary to indicate what spikes in fig. 2a the authors are addressing to have been remained in fig. 3a. Does fig 2b and 3b belong to the sample sections in fig 2a and 3a, respectively, or do they belong to the entire EEG set and Figs 2a and 3a are example sections of the entire data (with potentially not shown spectral representations)?

Section 4.2:

The claim in lines 277-279 “It is worth noting … many spikes” needs to be backed up by representing some results/demonstration. Line 282: The claim of “seven intrinsic mode functions were found to be the best trade off” needs a robust mathematical justification (i.e., sampling frequency related issues etc etc). Line 285-287: authors claim that their proposed algorithm outperforms the empirical mode decomposition method in suppressing the spikes, however, from the shown example, it is clear that their proposed algorithm adds distortion to data.

Reviewer 3 Report

The authors introduced a hybrid noise cancellation technique based on SSA and LRD. The method loos promising but the paper lacks proper experimental designs. My detailed comments are as follows:   What does ideal lowpass filter mean in line 14. How much your performance could the improve in performance? there is no quantitative performance comparison in the paper. What is z in line 102. Figure 1 is hear to read. As authors are using SSA why the initial band pass filter is needed? SSA can easily be used for bandpass filtering as well. I think in line 175 it should be greater than threshold not less than There is no description of the data provided. Where is the spikes? How do you know they are the spikes you are interested to remove? what is the noise level? there is not enough experiments in the paper. What is the impact of window length or noise level on the performance? How the threshold is selected in the method? Figure 5 needs more description.

Reviewer 4 Report

This paper works on removing the spike in EEG signal by presenting an iterative joint singular spectrum analysis and low rank decomposition. It is an incremental work by combining two existing models. The experimental results are promising in supporting the effectiveness of the presented model. Below are some comments for the authors’ consideration. 1) The process of how the EEG signal was collected is not mentioned in the paper. Additionally, the meaning of vertical axis of Figure 2a) is not clear, that is, what is the unit of 850? The authors need to clarify this point. 2) In Figure 5, it is not clear how these subfigures were obtained. The authors only mentioned the summed singular spectrum analysis. 3) The central idea is iteratively using the singular spectrum analysis and low rank decomposition to remove the spike in EEG signal. Therefore, it is of great necessity to give the intermediate process of the proposed model. That is, I want to see how the spike was the gradually removed in the evolution of the proposed model. 4) The following article is closely related to the low rank modeling in EEG signal processing. It would be grateful if the authors could include them in the reference list. Wanzeng Kong, Xianghao Kong, Qiaonan Fan, Qibin Zhao, Andrzej Cichocki: Task-free brainprint recognition based on low-rank and sparse decomposition model. IJDMB 22(3): 280-300 (2019) Xianghao Kong, Wanzeng Kong, Qiaonan Fan, Qibin Zhao, Andrzej Cichocki: Task-Independent EEG Identification via Low-Rank Matrix Decomposition. BIBM 2018: 412-419

Round 2

Reviewer 1 Report

Corrections made, based on the comments, are satisfactory.

Still, English need to be  improved. You are not using correctly  the plural tense. Often in the text you use the term "signal spike corrupted".To my understanding this should read "Signal spike corruption". Is that  so or you mean something else?

Reviewer 2 Report

This reviewer acknowledges the amendments to the paper.

- However, I would highly recommend using a unique graphical dimension-size (width-length) for the figures with similar x/y axis lengths (either in spectral or time domain). This will make it much easier for the reader to follow/comprehend.

As an example, Fig 2a and Fig. 3 and 4a are of the same numerical length, but they are printed with different lengths on the MS. This suggestion is also applicable to Fig.7, Fig.8 and 9 and so on.

- In addition, details on choosing the value of many of the parameters have to be explained clearly and yet remains unclear. This reviewer suggests adding such info to strength up the MS.

Reviewer 3 Report

Some of my concerns are still missing in the authors response or it is hard to follow:

Data description does not include the information regarding the application, sampling rate, appearance of spikes and their location; Threshold selection is still confusing; the threshold was selected by on looking at the signals and then set the threshold. How are you sure the selected threshold is extendable to other signals? Do you need to select a threshold for each new signal? I would suggest authors to simulate a new signal with different spikes and see if the same behaviour can be seen. As the signal was simulated, various noise levels could be added to check the impact of noise. 

Round 3

Reviewer 3 Report

Authors answered all my issues.